# PCH1 and PCHL promote photomorphogenesis in plants by controlling phytochrome B dark reversion

Beatrix Enderle[1], David J. Sheerin[1], Inyup Paik[2], Praveen Kumar Kathare[2], Philipp Schwenk[1,3], Cornelia Klose [1], Maximilian H. Ulbrich[3,4,5], Enamul Huq [2] & Andreas Hiltbrunner [1,3,5]

Phytochrome B (phyB) is the primary red light photoreceptor in plants, and regulates both growth and development. The relative levels of phyB in the active state are determined by the light conditions, such as direct sunlight or shade, but are also affected by light-independent dark reversion. Dark reversion is a temperature-dependent thermal relaxation process, by which phyB reverts from the active to the inactive state. Here, we show that the homologous phyB-binding proteins PCH1 and PCHL suppress phyB dark reversion, resulting in plants with dramatically enhanced light sensitivity. Moreover, far-red and blue light upregulate the expression of *PCH1* and *PCHL* in a phyB independent manner, thereby increasing the response to red light perceived by phyB. PCH1 and PCHL therefore provide a node for the molecular integration of different light qualities by regulation of phyB dark reversion, allowing plants to adapt growth and development to the ambient environment.

[1] Faculty of Biology, Institute of Biology II, University of Freiburg, 79104 Freiburg, Germany. [2] Department of Molecular Biosciences and The Institute for Cellular and Molecular Biology, The University of Texas at Austin, Austin, TX 78712, USA. [3] Spemann Graduate School of Biology and Medicine (SGBM), University of Freiburg, 79104 Freiburg, Germany. [4] Department of Medicine, Renal Division, Freiburg University Medical Center, 79106 Freiburg, Germany. [5] BIOSS Centre for Biological Signalling Studies, University of Freiburg, 79104 Freiburg, Germany. Correspondence and requests for materials should be addressed to A.H. (email: andreas.hiltbrunner@biologie.uni-freiburg.de)

Phytochromes are red/far-red photoreceptors in plants that play a critical role in the adaptation of growth and development to the environment[1–3]. Gene duplication events during evolution of seed plants resulted in small gene families coding for different types of phytochromes, including phytochromes A–E (phyA–E) in *Arabidopsis*[4]. Phytochrome A (phyA) and phytochrome B (phyB) play a dominant role in seed plants[2]. PhyA is required for sensing far-red light (FR) and weak light of any wavelength, while phyB is the primary receptor for red light (R). Phytochromes exist in two states, an inactive Pr and a biologically active Pfr form that have absorption peaks in R (660 nm) and FR (730 nm), respectively. Pr and Pfr reversibly convert into each other upon absorption of light, resulting in high levels of the active Pfr form in R and low levels in FR[5]. In addition to light-induced Pfr → Pr reversion, the active Pfr state can also revert to the inactive Pr state in a light-independent thermal relaxation process referred to as dark reversion. The physiological activity of phyB, the primary phytochrome in light-grown and adult plants, is strongly affected by dark reversion[6–8]. As such, dark reversion is a major factor that determines how plants respond to light. Several phyB mutant alleles with altered dark reversion have been discovered, but despite the strong impact of dark reversion on the activity of phyB, no extragenic mutants with altered phyB dark reversion had been identified yet[7–12]. It is also unknown, whether plants actively regulate dark reversion of phyB to control photomorphogenic growth and development.

Recently, PHOTOPERIODIC CONTROL OF HYPOCOTYL 1 (PCH1) has been identified as a protein binding to light-activated phyB[13,14]. Seedlings lacking functional PCH1 have elongated hypocotyls compared to the wild type when grown in short days[14]. Under these conditions, PCH1 transcript and protein levels peak at dusk, enhancing phyB-dependent inactivation of the growth-promoting transcription factor PHYTOCHROME INTERACTING FACTOR 4 (PIF4)[13,14]. In contrast, PCH1 levels are low towards the end of the night, leading to increased PIF4 activity and hypocotyl growth[14]. Thus, PCH1 has been suggested to integrate clock and light signals through modulation of diurnal phyB activity. However, PCH1 is a protein of unknown molecular function and the mechanism by which it enhances phyB-dependent light responses is still unknown.

Here, we show that the *pch1 pchl* double mutant, which lacks functional PCH1 and a homologue, PCH1-LIKE (PCHL), displays strongly accelerated phyB dark reversion. Moreover, we show that additional signalling pathways control the expression of *PCH1* and *PCHL*, thereby affecting the activity of phyB. Thus, PCH1 and PCHL allow plants to adapt to their environment by acting as a node of signal integration that regulates phyB dark reversion.

## Results

**PCH1 and PCHL interact with light-activated phyB**. Using two independent approaches, yeast two-hybrid (Y2H) screening and MS/MS analyses of immunoprecipitated phyB complexes from extracts of phyB-NLS-GFP expressing *Arabidopsis* plants, we identified PCH1 as a phyB interacting protein (Fig. 1a, b; Supplementary Fig. 1a). PCH1 has previously been identified as a phyB interaction partner[13,14]. Based on homology searches, we found one potential homologue of PCH1 in *Arabidopsis*, PCH1-

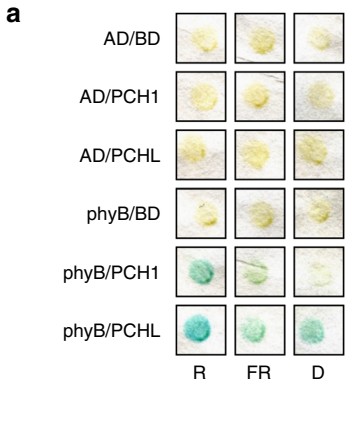

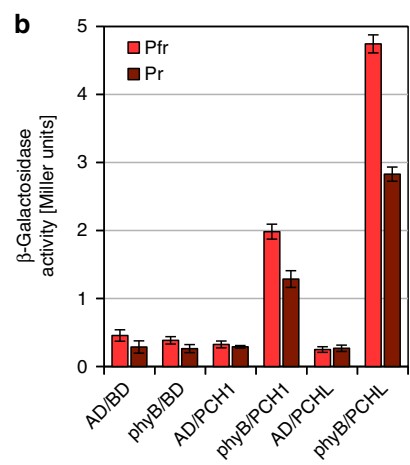

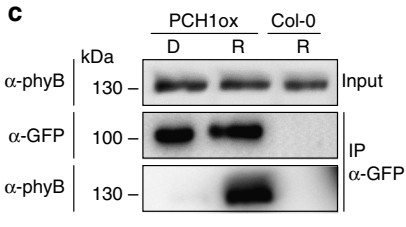

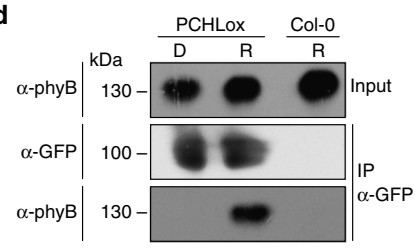

**Fig. 1** Light-activated phyB interacts with PCH1 and PCHL. **a**, **b** Y2H detection of interaction of phyB with PCH1 and PCHL. The phyB-GAL4-activation domain (phyB-AD) fusion was co-expressed with GAL4-DNA-binding domain (BD-) fusions of PCH1 or PCHL. **a** X-Gal filter-lift assay. Yeast cells were lifted from chromophore-supplemented plates pre-incubated for 48 h in either constant darkness (D), red (R), or far-red light (FR). **b** ONPG assay. Yeast cultures supplemented with chromophore were exposed to red light, to convert phyB to Pfr, or to far-red light for phyB Pr. $n = 3$; error bars indicate $\pm$ s.e.m. **c**, **d** PCH1 and PCHL co-purify with phyB from native plant extracts. Four-day-old dark-grown seedlings over-expressing HA-YFP-PCH1 (PCH1ox) **c** or HA-YFP-PCHL (PCHLox) **d** in the Col-0 wild type background were either treated with red light (R, 7 µmol m$^{-2}$ s$^{-1}$) for 10 min or kept in darkness (D). Native total protein extracts were prepared and used for co-immunoprecipitation (Co-IP) assays. IP was performed using α-GFP antibody. α-phyB and α-GFP antibodies were used to detect endogenous phyB and YFP-tagged PCH1 and PCHL, respectively

LIKE (PCHL) that was not identified in the previous reports (Supplementary Fig. 2). PCHL also interacts with phyB in the Y2H system (Fig. 1a, b); moreover, interaction of phyB with PCH1 and PCHL is confirmed by co-immunoprecipitation (Co-IP) from stable transgenic *Arabidopsis* plants (Fig. 1c, d; Supplementary Fig. 1b), as well as from mammalian cells (Supplementary Fig. 1c, d). Co-IP assays from plants suggest that PCH1 and PCHL preferentially interact with phyB Pfr, and a weak preference for binding to Pfr was also observed in the Y2H system.

**PCH1 and PCHL extend phyB activity into the dark phase**. Although PCH1 was previously shown as a phyB interactor, the molecular mechanism by which it promotes phyB-mediated light responses remained elusive[14]. Seedlings over-expressing HA-Yellow Fluorescent Protein (YFP)-tagged PCH1 (PCH1ox) or PCHL (PCHLox) are strongly hypersensitive to continuous R (Supplementary Fig. 3a–e), while seedlings lacking functional PCH1 and PCHL are hyposensitive (Supplementary Fig. 3f); PCH1 and PCHL have only a weak effect on hypocotyl growth in FR, consistent with a function of PCH1 and PCHL in phyB signalling (Supplementary Fig. 3g, h)[14]. Dark-grown PCH1ox and PCHLox seedlings are fully etiolated but we observed slightly reduced hypocotyl elongation in the dark compared to the wild type (Fig. 2a, b; Supplementary Fig. 4). Interestingly, we found that PCH1ox and PCHLox seedlings also respond to daily R pulses similar to seedlings expressing phyB mutant versions with reduced dark reversion, exhibiting short hypocotyls and open cotyledons (Fig. 2a, b; Supplementary Fig. 4)[11]. In contrast, wild type Col-0 seedlings are insensitive to R pulses and the response in seedlings over-expressing wild type phyB is very weak due to rapid phyB Pfr→Pr dark reversion between light pulses[11]; therefore, we hypothesised that PCH1 and PCHL might increase the levels of phyB Pfr by inhibiting phyB dark reversion. The total amount of phyB is critical for the responsiveness to R light[15], but immunoblot analyses confirm that the amount of phyB is similar in wild type, PCH1ox, PCHLox, *pch1*, *pchl*, and *pch1 pchl* backgrounds (Supplementary Fig. 5). If PCH1 and PCHL inhibit phyB dark reversion but not Pfr→Pr photoconversion, then a long-wavelength FR pulse immediately following a R pulse would be expected to revert the effects of the R pulse. This was indeed the case (Fig. 2a, b; Supplementary Fig. 4), suggesting that over-expression of PCH1 and PCHL does not affect the intrinsic photoreversibility of phyB.

To further investigate the possibility that PCH1 and PCHL allow maintenance of high levels of phyB Pfr in the dark, we grew wild type, PCH1ox, and PCHLox seedlings in 8 h R/16 h dark cycles and treated them with FR pulses either at the end of the day or 4, 8, or 12 h after light-off to trigger Pfr→Pr reversion (Fig. 2c). Hypocotyl growth in PCH1ox and PCHLox seedlings was more strongly affected than in the wild type by the reverting effect of FR pulses applied during the dark phase, suggesting that higher levels of photoreversible phyB Pfr have been maintained in the dark in seedlings over-expressing PCH1 or PCHL than in wild type seedlings. These data are consistent with the hypothesis that PCH1 and PCHL reduce phyB dark reversion.

To investigate the effect of endogenous levels of PCH1 and PCHL on the activity of phyB we used *pch1*, *pchl*, and *pch1 pchl* mutant seedlings for an end-of-day far-red (EOD-FR) experiment (Fig. 2d; Supplementary Fig. 6). Seedlings were grown in 8 h R/16 h dark cycles with or without a FR pulse at the end of the light phase. Wild type seedlings respond to a FR pulse at light-off with increased hypocotyl growth, which is due to phyB Pfr→Pr photoconversion[16]. In contrast, *pch1* and *pch1 pchl* seedlings not treated with a FR pulse grew as tall as those exposed to a FR pulse

at the end of the light phase, consistent with rapid loss of active phyB after light-off due to dark reversion. Seedlings lacking functional PCHL also exhibited increased hypocotyl growth in light/dark cycles compared to continuous light but were partially responsive to EOD-FR pulses. These data suggest that PCH1 and PCHL are necessary for robust phyB activity in vivo, possibly by reducing dark reversion of phyB and thereby stabilising phyB Pfr in darkness.

Light-activated phyB, and many phyB-binding proteins, assemble into subnuclear structures called photobodies[17]. Co-expression of phyB-mCerulean (mCer) and HA-YFP-PCH1 or HA-YFP-PCHL revealed that PCH1 and PCHL also co-localise with phyB in light-induced photobodies in stable transgenic *Arabidopsis* (Fig. 2e; Supplementary Fig. 7a). Only phyB Pfr is retained in photobodies, while phyB Pr rapidly dissociates[11,18]. Co-expression of PCH1 strongly accelerated the appearance of phyB-mCer photobodies after light-on and prevented dissociation after light-off, similar to previous observations (Fig. 2e; Supplementary Fig. 7b)[14]. Over-expression of PCHL had a qualitatively similar but weaker effect on the stability of phyB-mCer photobodies, consistent with the hypothesis that PCH1 and PCHL promote retention of phyB Pfr in darkness. Moreover, phyB photobodies were highly unstable in the *pch1 pchl* double mutant (Fig. 2f). Importantly, over-expression of PCH1 did not prevent the dissociation of phyB-mCer photobodies in response to a FR pulse given at light-off (Supplementary Fig. 7c), further supporting the hypothesis that PCH1 and PCHL reduce phyB dark reversion but do not affect phyB Pfr → Pr photoconversion.

**PCH1 and PCHL inhibit phyB dark reversion**. To directly measure the effect of PCH1 and PCHL on the in vivo ratio between active and total phyB (Pfr/Ptot), we utilised a dual-wavelength ratiospectrophotometer (hereafter referred to as a ratiospect). Only endogenous phyA levels are sufficiently high in wild type seedlings to be detected by ratiospect measurements, but phyB Pfr/Ptot levels can be measured in seedlings over-expressing phyB-GFP—either in the *phyA* mutant background or following a 3 h R pre-irradiation to trigger degradation of phyA[11]. After saturating R treatment and transfer to the dark, phyB Pfr/Ptot levels dropped more rapidly in the *pch1 pchl* double mutant background than in the wild type, whilst the total amount of phyB did not change (Fig. 3a, b; Supplementary Fig. 8). Moreover, co-expression of c-Myc-mCherry-PCH1 or c-Myc-mCherry-PCHL in phyB-GFP over-expressing seedlings strongly reduced phyB dark reversion following pre-irradiation with R for either 20 min or 3 h (Fig. 3c, d). Intriguingly, phyB dark reversion in PCH1ox and PCHLox seedlings was substantially slower following the 3 h R pre-irradiation in comparison to only 20 min pre-irradiation, despite both treatments establishing equal initial phyB Pfr/Ptot levels. This observation could be explained through increased phyB nuclear transport, and thus binding to the nuclear localised PCH1 and PCHL. Alternatively, more efficient assembly of phyB into photobodies that are known to form after prolonged exposure to R and to reduce phyB dark reversion could also explain the increased Pfr stability[11,18–20].

**PCH1 and PCHL function in molecular signal integration**. Signal integration is critical for the adaptation of growth and development of plants to diverse and rapidly changing environmental conditions. Given that PCH1 and PCHL regulate the activity of phyB, they may form a node for input for other signalling pathways upon responses to R. Using qPCR analyses we found that both *PCH1* and *PCHL* expression in dark-grown seedlings is transiently upregulated following transition to R, whilst transition to FR or blue light (B) results in sustained

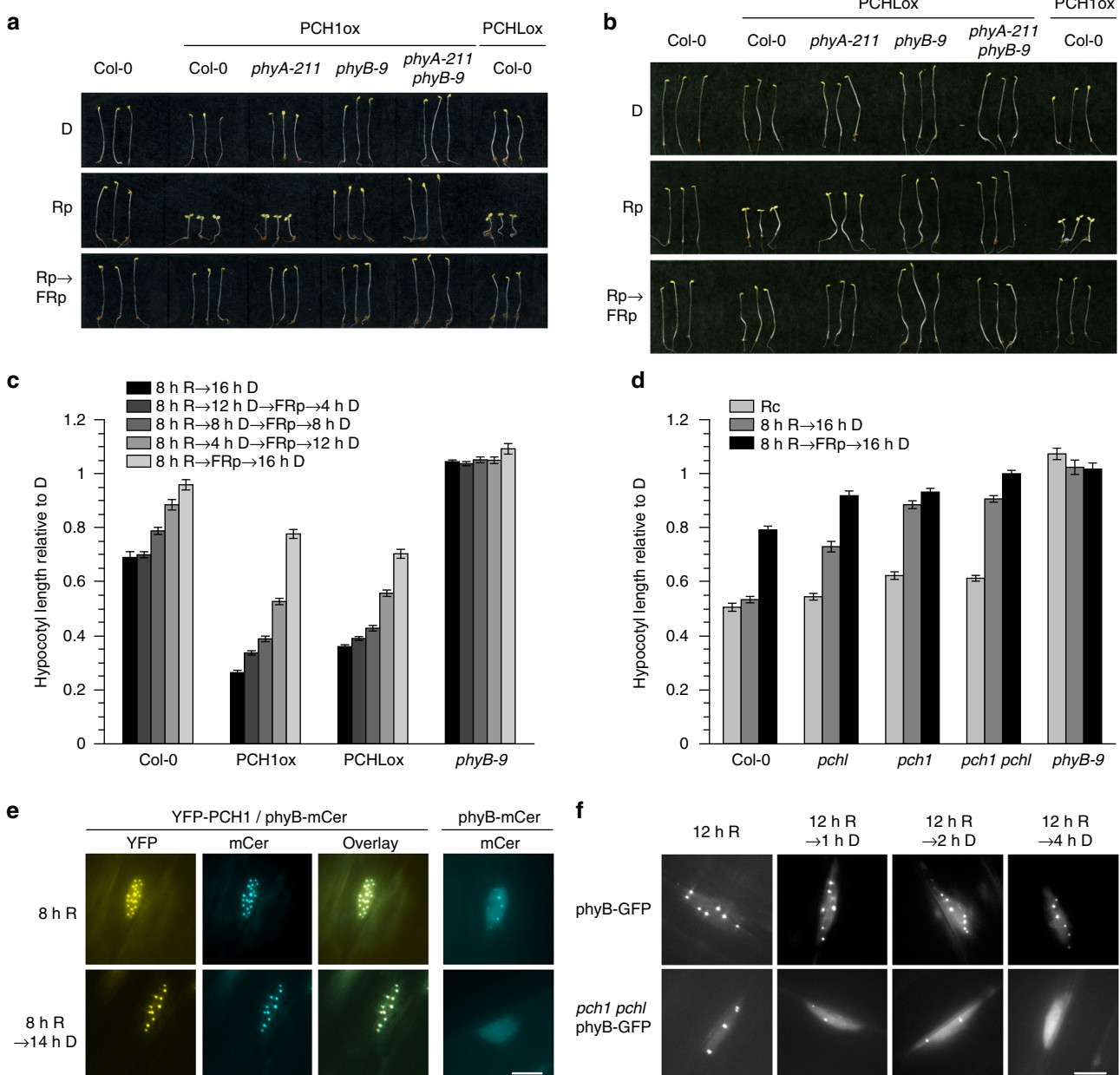

**Fig. 2** PCH1 and PCHL stabilise phyB in the active state in planta. **a**, **b** PCH1ox and PCHLox seedlings respond to red light pulse treatments (Rp). Wild type (Col-0) and mutant seedlings expressing either HA-YFP-PCH1 (PCH1ox) (**a**) or HA-YFP-PCHL (PCHLox) (**b**) were grown for 4 days in darkness on filter paper soaked with water. The seedlings were either treated with a single red light pulse (Rp, 5 min, 50 µmol m$^{-2}$ s$^{-1}$) per day, or a Rp followed by a long-wavelength FR pulse (FRp, 776 nm, 5 min, 50 µmol m$^{-2}$ s$^{-1}$) (Rp → FRp). Control seedlings were kept in darkness (D). See Supplementary Fig. 4 for quantification of hypocotyl lengths and experiments with seedlings grown on 0.5× MS medium. **c** High levels of active phyB are maintained during the dark phase in PCH1ox and PCHLox seedlings. Wild type (Col-0), PCH1ox, PCHLox, and *phyB-9* seedlings were grown for 4 days in 8 h red (R, 50 µmol m$^{-2}$ s$^{-1}$)/16 h dark (D) cycles and given a long-wavelength far-red light pulse (FRp, 776 nm, 5 min, 50 µmol m$^{-2}$ s$^{-1}$) at time points after lights-off. Control seedlings were kept in darkness (D). **d** The end-of-day far-red (EOD-FR) response requires PCH1 and PCHL. Wild type (Col-0) and mutant seedlings were grown as in **c**, except either constant red light (Rc), an immediate far-red light pulse (8 h R → FRp → 16 h D), or no far-red light pulse (8 h R → 16 h D) were used. **c**, **d** Mean hypocotyl length relative to dark-grown seedlings is shown. Error bars indicate ± s.e.m.; $n \geq 20$. **e**, **f** Subnuclear localisation of PCH1 and phyB was analysed by fluorescence microscopy. Scale bar = 5 µm. **e** PCH1 stabilises phyB photobodies. Four-day-old etiolated seedlings expressing phyB-mCer in *phyB-9* or HA-YFP-PCH1 (PCH1ox) backgrounds were exposed to red light (R, 10 µmol m$^{-2}$ s$^{-1}$) for 8 h, followed either by 0 or 14 h incubation in darkness (D). Data for phyB-mCer single transgenic seedlings are duplicated in Supplementary Fig. 7a. **f** PhyB photobodies are highly unstable in the *pch1 pchl* mutant. Four-day-old etiolated seedlings expressing phyB-GFP in wild type or *pch1 pchl* backgrounds were exposed to red light (R, 50 µmol m$^{-2}$ s$^{-1}$) for 12 h followed by incubation in darkness (D)

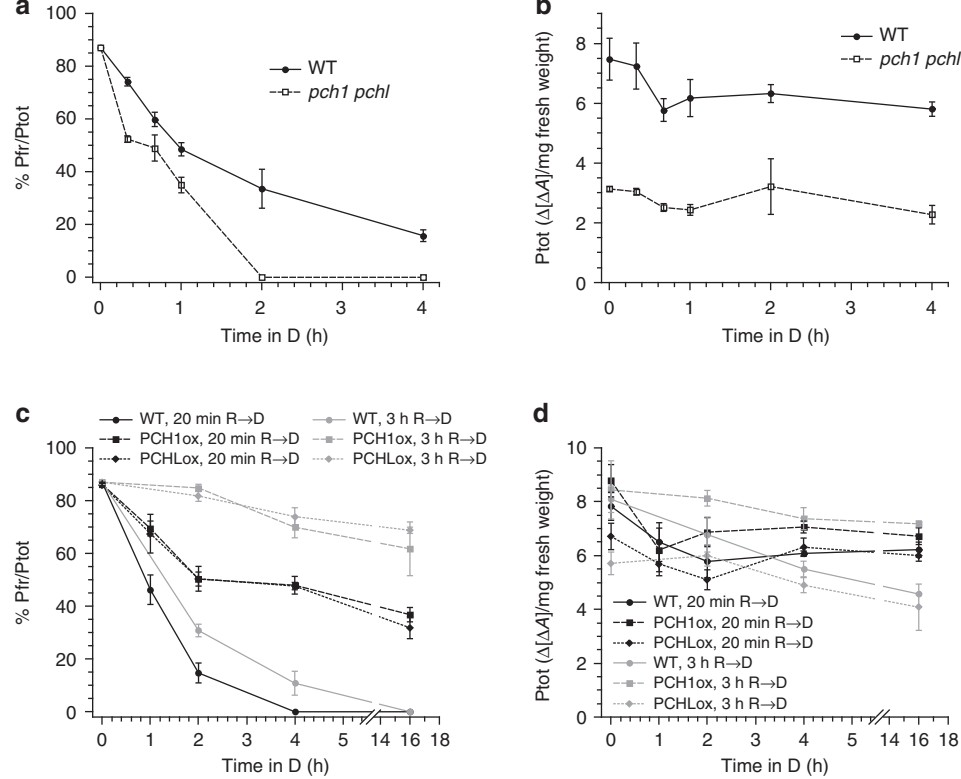

**Fig. 3** PCH1 and PCHL inhibit phyB dark reversion. Dual-wavelength ratiospectrophotometer quantification of the relative abundance of phyB in the active state (Pfr/Ptot) (**a**, **c**) and total phyB (Ptot) (**b**, **d**) in planta. **a**, **b** PhyB dark reversion is accelerated in the absence of PCH1 and PCHL. Four-day-old etiolated seedlings expressing phyB-GFP in PCH1/PCHL wild type (WT; phyA-211 phyB-9 phyB-GFP) or the mutant background (pch1 pchl; phyA-211 pch1 pchl phyB-GFP) were exposed to red light (R, 10 μmol m$^{-2}$ s$^{-1}$) for 3 h followed by incubation in darkness (D) for indicated times. **c**, **d** PhyB dark reversion is reduced in PCH1ox and PCHLox seedlings. Four-day-old etiolated seedlings expressing c-Myc-mCherry-PCH1 (PCH1ox) or c-Myc-mCherry-PCHL (PCHLox) in the phyB-GFP background were pre-irradiated with red light (R, 20 μmol m$^{-2}$ s$^{-1}$) for either 20 min (black lines and symbols) or 3 h (grey lines and symbols) followed by incubation in darkness (D). **a–d** Data are means of ≥ 4 biological replicates; error bars indicate ±s.e.m.

upregulation of PCH1 and PCHL transcript levels (Supplementary Fig. 9). This expression pattern is also observed in PCH1 and PCHL promoter-GUS reporter lines, showing PCH1 and PCHL promoter activity primarily in cotyledons and parts of the hypocotyl, and for PCH1 also in roots (Supplementary Fig. 10a). In 14-day-old plants grown in either long-day or short-day conditions, PCH1 and PCHL promoter activity is detected in leaves, and for PCH1 in roots (Supplementary Fig. 10b). Interestingly, PCH1 promoter activity in roots appears not to be regulated by light. Using pch1 seedlings complemented with pPCH1:HA-YFP-PCH1 (Supplementary Fig. 3i), we also confirmed light-induced accumulation of PCH1 protein in FR, B, and R (Fig. 4a, b). Importantly, light regulation of PCH1 and PCHL transcript levels does not require phyB but depends on phyA in R and FR and on both phyA and cryptochromes in B (Fig. 4c, d; Supplementary Figs. 11 and 12). The phyA protein is degraded in R with a half-life of about 30 min, while degradation is much slower in FR[21]. Thus, rapid degradation of phyA in R might be the reason why upregulation of PCH1 and PCHL in R is only transient. We hypothesise that FR and B signalling may integrate with phyB signalling through PCH1 and PCHL, given that the expression of PCH1 and PCHL is upregulated in response to FR and B.

The responsiveness of etiolated wild type seedlings to R pulses has been reported to be strongly increased by pre-treatments with FR or B for 8 h[22,23]. In contrast, we found that PCH1ox and PCHLox seedlings are sensitive to R pulses even without FR or B pre-irradiation (Fig. 2a, b; Supplementary Fig. 4), while R pulses did not further promote hypocotyl growth inhibition and

cotyledon unfolding in phyA-211 and pch1 pchl mutant seedlings pre-treated with FR or B (Fig. 4e, f; Supplementary Fig. 13). Thus, induction of PCH1 and PCHL expression triggered through FR-induced and B-induced phyA signalling, can enhance phyB-mediated R responses by suppressing phyB dark reversion and thereby preventing inactivation of phyB Pfr (Fig. 5). It is tempting to speculate that also other signalling pathways could control the activity of phyB through regulation of PCH1 and PCHL, which might be critical for the adaptation of plants to changes in the ambient environment.

## Discussion

Dark reversion experiments performed for phyB expressed in E. coli, yeast and Arabidopsis demonstrate that dark reversion can be extensively modulated by external factors although it is an intrinsic property of the phyB molecule[8,18,24–26]. Here, we provide genetic, photo-biological and biochemical evidence that PCH1 and PCHL suppress phyB dark reversion. Importantly, the effect of PCH1 and PCHL on phyB dark reversion does not depend on over-expression; we show that endogenous levels of PCH1 and PCHL are sufficient to control the activity of phyB through regulation of dark reversion, establishing PCH1 and PCHL as key players in photomorphogenesis and signal integration.

Two fundamentally different models can be envisaged to explain how PCH1 and PCHL suppress dark reversion of phyB. PCH1 and PCHL could either directly stabilise the Pfr conformation of phyB to prevent reversion to the inactive Pr state or

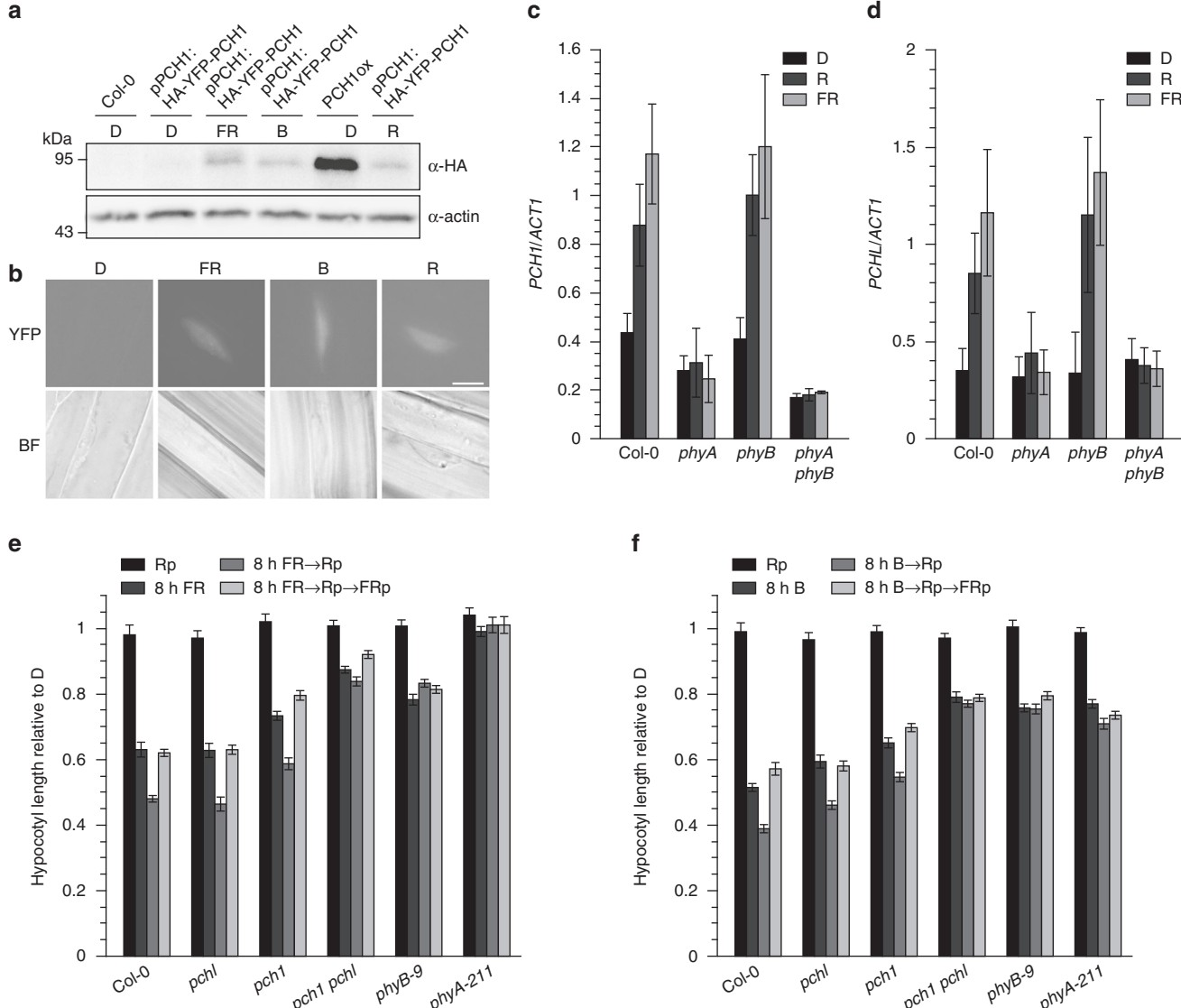

**Fig. 4** PCH1 and PCHL contribute to molecular signal integration. **a**, **b** PCH1 protein accumulates in far-red and blue light. **a** Four-day-old etiolated *pch1* seedlings expressing HA-YFP-PCH1 from its native promoter were either kept in darkness (D), or exposed to far-red (FR, 25 μmol m$^{-2}$ s$^{-1}$), blue light (B, 25 μmol m$^{-2}$ s$^{-1}$), or red light (R, 25 μmol m$^{-2}$ s$^{-1}$) for 8 h and harvested. Etiolated HA-YFP-PCH1 over-expressing seedlings (PCH1ox) were used for comparison. Total protein was extracted and used for immunoblotting; α-HA and α-actin antibodies were used for detection of HA-YFP-PCH1 and actin (loading control). **b** *pch1* seedlings expressing HA-YFP-PCH1 from its native promoter were grown as described in **a** and used for fluorescence microscopy. Scale bar = 5 μm. **c**, **d** Regulation of *PCH1* and *PCHL* expression by red and far-red light requires phyA. Four-day-old dark-grown wild type (Col-0) and mutant seedlings were either kept in darkness (D) or exposed to red (R, 7.5 μmol m$^{-2}$ s$^{-1}$) or far-red light (FR, 6.7 μmol m$^{-2}$ s$^{-1}$) for 1 h. Total RNA was extracted and qRT-PCR was performed using probes specific for either *PCH1* (**c**) or *PCHL* (**d**). *ACT1* was used as internal control and expression of *PCH1* and *PCHL* is shown relative to expression of *ACT1*. Data are means of three biological replicates; error bars indicate ± s.d. Biological replicates are shown in Supplementary Fig. 11; data in **c** and **d** correspond to Supplementary Fig. 11d and h. **e**, **f** *PCH1* and *PCHL* are required for phyB responsiveness amplification by far-red and blue light. Dark-grown wild type (Col-0) and mutant seedlings were pre-treated with either far-red (FR, 25 μmol m$^{-2}$ s$^{-1}$) (**e**) or blue light (B, 25 μmol m$^{-2}$ s$^{-1}$) (**f**) and given a red light pulse (Rp, 5 min, 50 μmol m$^{-2}$ s$^{-1}$), either followed by a long-wavelength far-red light pulse (FRp, 776 nm, 5 min, 50 μmol m$^{-2}$ s$^{-1}$) or darkness; these light pulses were repeated after 24 h. Mean hypocotyl length relative to dark-grown seedlings is shown. Error bars indicate ± s.e.m.; n ≥ 20. See Supplementary Fig. 13 for seedling photographs and detailed explanation of light treatments

they could induce molecular events that indirectly lead to stabilisation of phyB in the Pfr state. Phosphorylation at serine residue 86, for instance, accelerates phyB dark reversion[8] and PCH1/PCHL hypothetically could either prevent phosphorylation or promote dephosphorylation. It has also been demonstrated that phyB bound to photobodies is protected from dark reversion[18]; therefore, another hypothetical mechanism for the PCH1/PCHL-mediated suppression of phyB dark reversion is that PCH1/PCHL promote the association of phyB with photobodies, where other factors could stabilise phyB in the Pfr state.

Over-expression of PCH1 or PCHL strongly suppress dark reversion of phyB. A similar but much weaker effect on phyB dark reversion has been observed for ARABIDOPSIS RESPONSE REGULATOR 4 (ARR4). Seedlings over-expressing ARR4 are slightly hypersensitive to R, possibly due to reduced phyB dark reversion. However, dark reversion has not been investigated in the *arr4* mutant; therefore, it is still unknown if endogenous levels of ARR4 are sufficient to regulate dark reversion of phyB[26,27]. This is in contrast to PCH1 and PCHL, which strongly suppress phyB dark reversion even when expressed at wild type levels. To

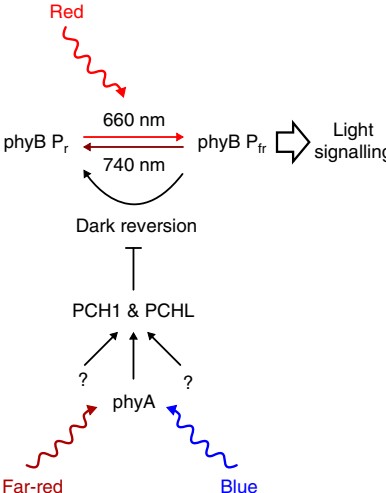

**Fig. 5** Model for PCH1-dependent and PCHL-dependent regulation of phyB responses. PCH1 and PCHL reduce phyB dark reversion. Blue and far-red light perceived by phyA induce expression of *PCH1* and *PCHL*, and thereby allow seedlings to respond to red light pulses (phyB responsiveness amplification by far-red and blue light). We speculate that also other signalling pathways could control the activity of phyB through regulation of PCH1 and PCHL, which might play an important role in integration of phyB-dependent light signalling with other signalling pathways

our knowledge, the *pch1 pchl* mutant is the first extragenic mutant shown to be affected in phyB dark reversion.

Though both PCH1 and PCHL delay phyB dark reversion, they do not contribute equally to physiological responses to light. It is possible that differences in expression patterns or light regulation of expression could be responsible for these differences. Furthermore, PCH1 and PCHL share only 32% identity in amino acid sequence and PCHL lacks two motifs conserved between PCH1 and sequences found in other species (corresponding to PCH1 amino acid residues 66–76 and 210–224), which could contribute to different biological activity.

Wild type seedlings are normally insensitive to daily R pulses but respond to this treatment upon pre-irradiation with FR or B[11,22,23]. This responsiveness amplification of phyB by FR and B depends on PCH1 and PCHL. Transcript levels of *PCH1* and *PCHL* are upregulated by phyA in response to FR and B, leading to suppression of phyB dark reversion and increased sensitivity to light (Fig. 5). Hypothetically, phyA-induced expression of *PCH1* and *PCHL* could further promote light responses by enhancing phyB downstream signalling, though experimental evidence is still lacking. The advantage of FR/B-dependent phyB responsiveness amplification for plants growing under natural conditions is unclear; however, this phenomenon demonstrates that plants actively control dark reversion of phyB to regulate the sensitivity to light.

We speculate that other signalling pathways could control PCH1 and PCHL and thereby integrate with phyB-mediated light responses to allow plants to adapt growth and development to the ambient environment. Moreover, phyB dark reversion is temperature-dependent and recent reports have shown that plants use this temperature dependency as a thermo-sensing mechanism[28,29]. Thus, PCH1 and PCHL could also play a role in integrating ambient light and temperature cues by controlling phyB dark reversion.

## Methods

**Vectors and cloning**. Details on the cloning of plasmid constructs and primers used in this study can be found in the Supplementary Note 1.

**Co-immunoprecipitation from mammalian cell culture**. HEK293T cells (provided by the BIOSS Toolbox, BIOSS Centre for Biological Signalling Studies, University of Freiburg), were cultivated in DMEM medium (PAN, Aidenbach, Germany) supplemented with 10% FCS (PAN, Aidenbach, Germany), 2 mM L-glutamine, 100 U/ml penicillin, and 0.1 mg/ml streptomycin (PAN, Aidenbach, Germany), at 37 °C, 5% CO₂, and high humidity. For transfection cells were seeded at 80–90% confluency in a 1:6 ratio in 10 cm tissue culture plates (Greiner Bio-One) the day prior transfection. Eight micrograms of total DNA was diluted in 560 μl of OptiMEM (Invitrogen), mixed with 24 μl of polyethylene–imine solution (1 mg/ml stock concentration; linear; MW: 25 kDa; Polyscience, Warrington, USA) and incubated for 15 min at room temperature before drop-wise addition to cells. Following 24 h, medium was replaced with fresh medium supplemented with 15 μM phycocyanobilin (Livchem #FSIP14137, Frankfurt/Main, Germany), and returned to incubate in darkness for a further 24 h prior to lysis.

The following steps were performed under green light. Cells were washed with 5 ml of DPBS (PAN, Aidenbach, Germany), and then lysed for 20 min on ice in 2.5 ml lysis buffer (100 mM NaPO₄ pH 7.8, 150 mM NaCl, 1 mM KCl, 1 mM EDTA, 1% (w/v) PEG4000, 50 μM MG132 (Sigma-Aldrich), 0.5% Triton-X100, 1 mM Na₃VO₄, 2 mM Na₄P₂O₇, 10 mM NaF, 1× protease inhibitor mixture (1 tablet per 50 ml; Roche)). Lysate was then centrifuged at 20,000×g for 20 min at 4 °C, and supernatant used immediately for co-immunoprecipitation.

Fifty microlitres of anti-GFP microbeads (Miltenyi Biotec, Bergisch-Gladbach, Germany; Cat-No: 130-091-125) were added to 2.5 ml of lysate, protected from light, and incubated at 4 °C for 2 h. Co-immunoprecipitation was performed following the procedure of the μMACS Anti-GFP Isolation Kit (Miltenyi Biotec, Bergisch-Gladbach, Germany; Cat-No: 130-091-125). Samples were separated by SDS-PAGE, and analysed by immunoblot using a monoclonal α-GFP antibody (Roche; Cat-No: 11814460001, monoclonal, mouse, dilution 1:2000) and a monoclonal α-c-Myc antibody (Abcam, Cambridge, UK; Cat-No: ab32, monoclonal, mouse, dilution 1:5000), respectively, and detected using ECLprime reagent (GE Healthcare).

**Plant material**. All *Arabidopsis thaliana* lines used were in Col-0 background. The *pch1* (T-DNA insertion line; SALK_024229), *phyA-211*, *phyB-9*, *cry1-304*, and *cry2-1* mutants have been described previously[14,30–34]; the *phyA-211 phyB-9* double mutant was generated by crossing the respective single mutants. The *pchl* mutant line contains a T-DNA insertion within the coding region of *PCHL* (AT4G34550) and was provided by the Nottingham Arabidopsis Stock Centre (SALK_206946C; #N696800). The *pch1 pchl* double mutant was generated by crossing the respective single mutants. Genotyping of *pch1* and *pchl* was performed using primers DS245/DS246 (wild type *PCH1*), DS246/LBb1.3 (T-DNA insertion in *pch1*), BE078/BE079 (wild type *PCHL*), and BE078/LBb1.3 (T-DNA insertion in *pchl*). The sequence of primers used for genotyping *pch1* and *pchl* can be found in the Supplementary Note 1.

The phyB-GFP over-expression line (p35S:PHYB-GFP) used for Pfr and dark reversion measurements in the dual-wavelength ratiospectrophotometer (ratiospect) (Fig. 3) is in *phyA-211 phyB-9* double mutant background and has been described previously[6]. This line was crossed to *pch1 pchl* to generate phyB-GFP *phyA-211 pch1 pchl*, which was used for ratiospect measurements (Fig. 3), Western blot quantification of phyB-GFP and endogenous phyB (Supplementary Fig. 8), and analysis of phyB photobodies in *pch1 pchl* (Fig. 2f). This line still contains endogenous phyB, which considerably contributes to total phyB levels and detection of phyB by ratiospect measurements (Supplementary Fig. 8).

The *Agrobacterium tumefaciens* floral dip transformation method[35,36] was used for generating PCH1ox and PCHLox lines. HA-YFP-PCH1ox (p35S:HA-YFP-PCH1:terRbcS; plasmid pDS366) and HA-YFP-PCHLox (p35S:HA-YFP-PCHL:terRbcS; plasmid pBE52c) were transformed into the Col-0 background, and subsequently crossed into *phyA-211*, *phyB-9*, and *phyA-211 phyB-9* backgrounds. These lines were used for all physiological experiments, the Co-IP assays shown in Fig. 1, and the Western blots in Supplementary Fig. 5.

PCH1ox (p35S:myc-mCherry-PCH1:terRbcS; plasmid pBE50) and PCHLox (p35S:myc-mCherry-PCHL:terRbcS; plasmid pBE60) were transformed into the above phyB-GFP line[6]. These lines were used for ratiospect measurements of Pfr levels (Fig. 3) and the Co-IP assay in Supplementary Fig. 1b.

For native level expression of PCH1, pPCH1:HA-YFP-PCH1 (pBE48) was transformed into the *pch1* line using the floral dip method[35,36]. These lines were used in Fig. 4a (line #2) and 4b (line #2), as well as in Supplementary Fig. 3i (lines #1 and #2).

A phyB-mCer (pPHYB:PHYB-mCer:terRbcS, pPPO43B-PHYB) line that over-expresses phyB was generated in the *phyB-9* background using the floral dip method[35,36], and crossed to PCH1ox *phyA-211* and PCHLox *phyA-211*. These lines were used for the experiments shown in Fig. 2e and Supplementary Fig. 7.

To generate phyB-NLS-GFP lines for purification of phyB complexes used for MS/MS analyses, pART27 p35S:PHYB-NLS-GFP was transformed into *phyA-211 phyB-9* double mutant background using the floral dip method[35]. Homozygous lines were selected on kanamycin for further experiments.

*PCH1* and *PCHL* promoter-GUS reporter constructs (plasmids pBE26new and pBE75) were transformed into Col-0 using the floral dip method[35,36]. Homozygous lines were selected by spraying T2 seedlings with BASTA. These lines were used for the experiments shown in Supplementary Fig. 10.

**Measurement of hypocotyl length**. Seedlings were grown in Petri dishes on four sheets of filter paper (Macherey-Nagel; Cat-No: MN 615) soaked with sterile water. Seeds were first stratified at 4 °C in darkness for 4–7 days, then germination induced by incubation in white light (70 µmol m$^{-2}$ s$^{-1}$) at 22 °C for 4–8 h. Following 16 h in darkness at 22 °C, seedlings were given a long-wavelength FR pulse (776 nm, 5 min, 50 µmol m$^{-2}$ s$^{-1}$), to inactivate all phytochromes before transfer to various conditions, which are described in figure legends. The hypocotyl length of at least 20 individual seedlings was measured at the fourth day following germination induction. Error bars represent the standard error of the population. Seedlings for the experiments shown in Supplementary Fig. 4d and f were grown in Petri dishes on four sheets of filter paper soaked with sterile 0.5× MS medium under the conditions described above.

**Histochemical GUS stain of *Arabidopsis* seedlings and plants**. Three-day-old dark-grown seedlings treated for 24 h with either R (660 nm, 25 µmol m$^{-2}$ s$^{-1}$), FR (740 nm, 25 µmol m$^{-2}$ s$^{-1}$), B (450 nm, 25 µmol m$^{-2}$ s$^{-1}$), or white light (100 µmol m$^{-2}$ s$^{-1}$), or 14-day-old seedlings grown in either long (16 h white light, 100 µmol m$^{-2}$ s$^{-1}$/8 h dark) or short day conditions (8 h white light, 100 µmol m$^{-2}$ s$^{-1}$/16 h dark) were submerged in 2 ml fixative solution (0.3 M mannitol, 10 mM MES, 0.3% formaldehyde, pH 5.6) for 45 min. The seedlings were then incubated in washing solution (50 mM Na$_2$HPO$_4$, pH 7) for 45 min on a shaker before transfer to the staining solution (50 mM Na$_2$HPO$_4$, 0.05% X-Gluc [Carl Roth, Germany], 2 mM K$_3$Fe(CN)$_6$, 2 mM K$_4$Fe(CN)$_6$, pH 7). Seedlings were vacuum infiltrated for 20 min and incubated over night at room temperature (22 °C). Then, seedlings were destained stepwise using a dilution series of 50%, 60% and finally 70% ethanol. Pictures were taken using a camera-equipped binocular microscope from Zeiss, Germany. Magnification was 6.4× for whole seedlings, 25.6× for leaves and 40× for roots.

**Quantification of transcript levels**. Four-day-old dark-grown seedlings were either kept in darkness or exposed to the respective light conditions for the indicated time. Total RNA was extracted using the concert plant RNA purification reagent (Invitrogen) followed by a RNA clean up using the RNAeasy kit (Qiagen). cDNA was prepared using the High Capacity Reverse Transcription kit (Applied Biosystems, Thermo Scientific). qRT-PCR was performed using Taqman probes specific for *PCH1*, *PCHL*, or *ACT1* (AT1G32200; internal control) and a 2× high ROX master mix (Eurogentec, Belgium). Relative *PCH1* or *PCHL* gene expression was calculated in comparison to *ACT1* expression. The sequence of primers and probes used for qRT-PCR can be found in the Supplementary Note 1.

**In vivo spectrometric analyses**. Dark reversion of phyB over-expression lines was measured in living seedlings in a dual-wavelength ratiospectrophotometer (ratiospect). Four-day-old dark-grown seedlings were irradiated for either 20 min or 3 h with saturating red light (20 µmol m$^{-2}$ s$^{-1}$), then returned to darkness (22 °C) for indicated times. Seedlings were transferred to ice and 100–140 mg of seedlings transferred to chilled ratiospect cuvettes. Ratiospect measurements were performed at 0.3–0.8 °C to avoid the effects of temperature. Each time point represents the average of at least four independent biological samples. The total amount of phytochrome (Ptot) and Pfr/Ptot values were calculated as previously described[37].

**Co-immunoprecipitation from *Arabidopsis***. Four-day-old dark-grown seedlings were either treated with red light (7 µmol m$^{-2}$ s$^{-1}$) for 10 min or kept in the dark. Immunopreciptiations were performed using IP buffer (50 mM Tris/HCl pH 7.5, 150 mM NaCl, 0.1% NP-40, 1× Protease inhibitor cocktail, 1 mM PMSF) and 1 µg α-GFP antibody (Abcam; Cat-No: ab6556, polyclonal, rabbit) bound to 20 µl Dynabeads (Life Technologies; Cat-No: 10002D). Following 1 h incubation in darkness, immunoprecipitated complexes were washed three times with IP buffer. Elutions were analysed by SDS-PAGE, and immunoblotting with α-phyB (monoclonal, mouse, B6-B3[38], dilution 1:250), α-c-Myc (Invitrogen; Cat-No: MA1-980, monoclonal, mouse, dilution 1:1000), and α-GFP (Invitrogen; Cat-No: A-11120, monoclonal, mouse, dilution 1:1000).

**Microscopy**. For epifluorescence microscopy seedlings were transferred to glass slides under dim green light. Epifluorescence microscopy was performed using an Axioplan microscope (Zeiss, Oberkochem, Germany) equipped with filter sets for GFP (Z13, excitation 470 nm, emission 493 nm; Zeiss), YFP (F31-028, excitation 500 nm, emission 515 nm; AHF Analysentechnik, Tübingen, Germany), and CFP (F31-044, excitation 436 nm, emission 455 nm; AHF Analysentechnik, Tübingen, Germany).

***Arabidopsis* protein extraction and immunoblot analysis**. Dark-grown seedlings were exposed to light as described in the figure legends. Total protein was extracted from harvested seedlings using preheated (95 °C) denaturing buffer (4 M Urea, 65 mM Tris/HCl pH 7.3, 3% SDS, 10% glycerol, 0.05% bromphenol blue, 10 mM DTT), 40–60 µg protein separated by 10% SDS-PAGE, and transferred to PVDF membranes. Membranes were probed with α-phyB antibody (monoclonal, mouse, B6-B3[38], 1:1000 dilution), α-HA-tag (Sigma-Aldrich; Cat-No: H3663, monoclonal,

mouse, 1:3000 dilution), α-actin (Sigma-Aldrich; Cat-No: A0480, monoclonal, mouse, 1:3000 dilution), and detected with alkaline phosphatase-coupled α-mouse IgG (Vector Labs, Burlingame, CA, USA; Cat-No: AP-2000, 1:10,000 dilution) using CDP-Star substrate (Roche Diagnostics) and a camera detection system (Fusion SL, Vilber-Lourmat, Germany). Quantification of immunblot signals in Supplementary Figs. 5 and 8 was done as previously described (Nicole Smalley; OpenWetWare—Protein Quantification Using ImageJ; http://www.openwetware.org/index.php?title=Protein_Quantification_Using_ImageJ&oldid=700558; 2017).

**Mass-spectrometry identification of PCH1**. Transgenic plants expressing phyB-NLS-GFP were germinated and grown in the dark for 4 days before irradiating for 10 min with red light (7 µmol m$^{-2}$ s$^{-1}$). Seedlings were harvested and ground in liquid nitrogen. Total protein was solubilised and immunoprecipitated in the IP buffer (50 mM Tris/HCl pH 7.5, 150 mM NaCl, 0.1% NP-40, 1× Protease inhibitor cocktail (Sigma-Aldrich, P9599), 1 mM PMSF) by 5 µg α-GFP antibody (Abcam; Cat-No: ab6556, polyclonal, rabbit) bound to 100 µl Dynabeads (Life Technologies; Cat-No: 10002D). Immunoprecipitation was performed with the same method as described in the Co-IP from plants. The collected beads were sent for liquid MS/MS analyses at JadeBio Inc., La Jolla, CA.

**Yeast two-hybrid methods**. Yeast two-hybrid assays were performed as previously described[39]. Briefly, yeast were cultured in or on media lacking leucine and tryptophan that was supplemented with 20 µM phycocyanobilin (Livchem #FSIP14137, Frankfurt/Main, Germany). Plates were incubated under constant red light (660 nm, 1.3 µmol m$^{-2}$ s$^{-1}$) or far-red light (740 nm, 13 µmol m$^{-2}$ s$^{-1}$) for 48 h at 26 °C for X-Gal filter lift assays. o-Nitrophenol-β-D-galactopyranoside assays were performed using yeast log-phase cultures exposed to either 10 min red or far-red light (7 µmol m$^{-2}$ s$^{-1}$).

**Image processing**. Uncropped gel and blot images are shown in Supplementary Figs. 14–22.

**Data availability**. The authors declare that all data supporting the findings of this study are available within the paper and its supplementary information files.

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

## Acknowledgements

This study was supported by the Excellence Initiative of the German Federal and State Governments (EXC 294/BIOSS to A.H. and M.H.U. and GSC-4/Spemann Graduate School to P.S.) and grants from the German Research Foundation (HI 1369/5-1 and HI 1369/7-1) to A.H., the Human Frontier Science Programme (HFSP; RGP0025/2013) to A.H. and E.H., and the National Institute of Health (NIH) (1R01 GM-114297) to E.H.; B. E. was supported by an LGFG (Landesgraduiertenförderung Baden-Württemberg) scholarship. We thank Prof. Jorge J. Casal (University of Buenos Aires) and Prof. Eberhard Schäfer (University of Freiburg) for discussion, Martina Krenz (University of Freiburg, Germany) for excellent technical work, and the BIOSS Toolbox for providing HEK293T cells.

## Author contributions

B.E. performed all physiological experiments, microscopy, ratiospect measurements, Y2H assays, and generated all transgenic lines and crossings; D.J.S. did the Y2H screen for phytochrome-interacting proteins; B.E. and P.K.K. performed qPCR experiments; I.P. did Co-IPs from plant extracts and MS/MS; P.S. performed Co-IPs from mammalian cells; C.K. did Western blot for endogenous phyB in *pch1 pchl*; M.H.U., E.H., and A.H. conceived the study; B.E., E.H., and A.H. wrote the manuscript; all authors discussed and commented on the manuscript.
