## [Peer Review File · Nature Communications]

Reviewers' comments:

Reviewer #1 (Remarks to the Author):

In this manuscript authors characterize the first two extragenic components, PHC1 and PHCL involved in phyB dark reversion. They conduct extensive photobiology experiments to demonstrate the involvement of PHC1 and PHCL in the control of phyB-dark reversion. While the involvement of PHC1 in light signaling is not new, authors here provide with solid evidence on the role of PHCL and PHC1 control dark reversion.

The topic is of interest and worth to be considered for publication. However clarification of a few points is requested at this point.

1) Authors describe the discovery of a PHC1 homologue, previously not identified. It will be of interest to provide a sequence comparison between PHC1 and PHCL. Are both F-box proteins? Do these genes have overlapping expression profiles?, are they expressed in the same developmental stage?

2) Being PHC1 and PHCL involved in dark reversion, a process tightly linked to temperature changes, have PHC1 and PHCL been tested for temperature regulation at the level of gene expression, protein accumulation and protein distribution?.

3) Authors state that the effect of PHC1 and PHCL over phyB dependent photomorphogenesis is not through regulation of phyB protein levels. However, SF4 protein blot may show slight differences of phyB in the dark. Quantification and replicas of the experiment would be required to clarify that no phyB protein changes are involved in phc1 and phcl phenotypes.

4) Figure 2F shows an apparent difference of phyB-GFP abundance in red light in the absence of phc1 phcl. Is this due to the optical plane used? Or are there differences in the abundance of phyB in R-light?

5) In Figure 4, authors show evidence of PHC1 and PHCL FR and B-light regulation at the level of gene expression. However, a missing piece of information is whether these changes in gene expression translate in changes in protein content and modulation of phyB-dark reversion. Can the authors complement their data with evidence of changes in protein content/distribution under the mentioned light regimes?

6) In their analyses of PHC1 and PHCL expression under R, B and FR(Figure S7) there are considerable differences in the kinetics and light sensitivity of PHC1 and PHCL expression. PHCL expression seems to be quite low under R regimes compared to B or FR, however the contribution to the phenotypes in Figures 10 and Figure 2 are similar. Can the authors comment on this observation? Is this regulation related to differential roles of PHC1 and PHCL?

7) In the same figure (S7), there are differences in the expression patterns of the 3 replicas provided. How does the average expression looks like? Are there kinetic differences on the expression of PHC1 and PHCL in response to light quality? (Fig S7).

8) Does the observed and FR light modulation of PHCL and PHC1 result in altered expression of phyB-signaling components? (i.e. PIFs, others?)
Is there an interaction of PHC1/L with phyA? ,and impact on phyA signaling under tested FR conditions?

9) PHC1 has been reported by other authors as an F-box protein, how would the potential function of PHC1/PHCL relate their involvement in dark-reversion?

Reviewer #2 (Remarks to the Author):

PCH1 was recently identified by tandem affinity purification coupled with mass spectrometry (AP-MS) analyses by the Nusinow group (Huang et al, eLife 2016). PCH1 has been demonstrated to interact directly with phyB, maintain phyB subnuclear photobodies in prolonged darkness, be regulated by the circadian clock, and mediate photoperiod-regulated seedling growth. This current study also independently identified PCH1 by both Y2H and immunoprecipitation coupled with MS/MS, highlighting the importance of PCH1 in the phyB signaling pathway. Their work focused more on extensive characterization of PCH1 and its homolog PCHL in stabilizing Pfr phyB and slowing phyB dark reversion process in darkness. Interestingly, they also found that photoactivated phyA by various wavelengths induces PCH1/PCHL transcript and protein levels, leading to enhanced phyB signaling, and thus revealing an integration point for the phyA action into phyB signaling networks. This work and previous publication collectively mark PCH1/PCHL as emerging key phyB signaling components, will inspire more researches from the plant photobiology community, especially from the perspectives of phyB as thermosensor and its application in crop breeding. My suggestions for improving the manuscript are listed below.

(1) Fig. 4A of the immunoblot assay, please add "R" control for comparison to FR and B light induction of HA-YFP-PCH1. It is suggestive to analyze PCHL protein levels in response to light treatments as well. Fig. 4B and S7A~S7C indicates ~3-fold difference in PCH1 transcript levels between dark and FR conditions, how come the difference in protein levels as shown in Fig. 4A and 4B is more than 3-fold (evidently detectable vs below detection threshold). In Fig. S7, two out of three biological replicates show that under R conditions, PCH1 is only transiently upregulated 1h after R exposure. It is well known that phyA is rapidly degraded upon R exposure; but within 1h, phyA is degraded by only ~50%; and by 3h, it is almost gone. Collectively, this suggests that R-induced PCH1 transcript upregulation is mediated by phyA before degradation. Such an interpretation is supported by Fig. 4C-D: in the phyA mutant, the PCH1/PCHL transcript level has no response to 1h R exposure. The conclusion drawn in the first paragraph of page 9 is thus well supported, but can be discussed more for its implication.

(2) It was not mentioned in the main text, but Fig 2A and 2B clearly show that in the dark PCH1ox and PCHLox are shorter than Col-0, phyB-9 and phyA-211phyB-9. Is it due to later germination of the transgenic lines, caused by the growth assay method (on filter paper plus water for growth; no MS salts), or indeed indicate the elongation suppression role of overexpressed PCH1/PCHL? Furthermore, the phenotype of PCH1/PCHLox causing hypersensitivity to Rp treatment is very striking. This observation came from growth assay on filter paper plus pure water. I would ask the authors to repeat the growth results on MS medium.

(3) In Fig S2, why the fluence rate response trends of Col-0 in panel D and F are so different from panels A~C, F and I, very minor change in hypocotyl length even under $\geq 20 \mu\text{mol m}^{-2} \text{s}^{-1}$ R. This is also different from most, if not all, reported Col-0 hypocotyl growth under continuous R. As such, the data of other genotypes in panels D-E are not so trustable. Please repeat the experiments.

(4) Fig. S4 western immunoblot assays indicate that phyB levels in Col-0 and pch1 pchl mutant are comparable, why total phyB measured in Fig. 3B by ratiospect is ~2-fold difference between Col-0 and the mutant, given that the same phyB-GFP OE line was used?

(5) For Fig. S5, please provide schematic illustration of the PCH1 and PCHL gene models, which should also include the T-DNA insertion sites and locations/orientations of primers used for transcript detection. I wonder if the primer pair used for the PCH1 transcript assay spans the T-DNA insertion site so that no transcript can be detected in the mutant.

(6) Fig. 1C, the immunoblot band intensities for the input proteins are so faint compared to the IP bands and input bands in Fig. 1D. Could the authors improve the blot image quality?

(7) The methods section is quite lengthy, especially about vectors and cloning, and plant materials. Meanwhile, the key information about the sequence lengths of promoters being cloned is not reported. The very detailed cloning methods as well as primer list can be put into supplemental data.

Minor points:

(1) Ref 45 misses the publication year.

(2) Page 19, line 381. Is the custom a-phyB antibody used for co-IP B6-B3, same as that used for immunoblot analysis (line 395)?

(3) Typo in Fig S1 legend, line 33; please swap "a-c-Myc" and "a-GFP" antibodies. Typo in Fig S10 legend, Line 137, "see Figures 4A and 4B for..." should be "see Figures 4E and 4F...".

RESPONSE TO REVIEWER 1:

Reviewer Comment:

1) Authors describe the discovery of a PHC1 homologue, previously not identified. It will be of interest to provide a sequence comparison between PHC1 and PCHL. Are both F-box proteins? Do these genes have overlapping expression profiles?, are they expressed in the same developmental stage?

Response:

- In the revised version of the manuscript we added a sequence alignment for *Arabidopsis* PCH1 and PCHL, and for PCH1/PCHL related sequences from other species (Supplementary Fig. 2).
- In contrast to PCH1, only one splicing variant of PCHL is annotated; this PCHL splicing variant is homologous to PCH1 splicing variant At2g16365.2 and does not contain an F-box motif. However, it is important to note that the PCH1 splicing variant At2g16365.2 that we used in our study and that has previously been used by Huang et al. (Huang et al., 2016, eLife, 5, 1–27) also does not contain an F-box motif. EST/transcript data (Phytozome jBrowse, TAIR gBrowse) provide only weak support for the existence of PCH1 splicing variants containing the F-box motif and no transcripts of splicing variants containing the F-box motif have been detected in other *Brassicaceae*, while transcripts coding for PCH1 variants not containing the F-box are present. Moreover, Huang et al. only found peptides mapping to the PCH1 variant not containing the F-box in their co-IP/mass spec approach used to identify PCH1 as a phyB and evening complex associated protein; they did not find any peptides mapping to the F-box containing version (Huang et al., 2016, eLife, 5, 1–27; Huang et al., 2016, Mol. Cell. Proteomics, 15, 201–217). Thus, it appears likely that the splicing variant At2g16365.2 (i.e. the variant not containing the F-box) is the most abundant version of PCH1 and that F-box containing versions may not exist or may not be expressed under the experimental conditions.
- In the revised version of the manuscript we include new experimental data on *PCH1* and *PCHL* expression obtained from promoter:GUS reporter lines (Supplementary Fig. 9). These data show that *PCH1* expression is high in the hypocotyl of dark-grown seedlings but very low in the cotyledons, while *PCHL* expression is generally low in the dark with weak promoter activity in the cotyledons and no activity in the hypocotyl. *PCH1* and *PCHL* expression is low upon 24 h exposure to red light, while expression – in particular in cotyledons – is strongly upregulated after 24 h in blue, far-red, or white light. This *PCH1/PCHL* expression pattern (low expression in D and after 24 h R; high expression after 24 h blue or far-red light) is consistent with the qRT-PCR data presented in Supplementary Fig. 8. *PCH1* promoter activity can also be detected in roots irrespective of the light conditions. In 14 day-old plants grown in either long or short day conditions, *PCH1* and *PCHL* are expressed primarily in leaves and *PCH1* also in the roots. We added a couple of lines in the revised version of the manuscript on the spatio-temporal regulation of *PCH1/PCHL* promoter activity (lines 166ff).

Reviewer Comment:

2) Being PHC1 and PCHL involved in dark reversion, a process tightly linked to temperature changes, have PHC1 and PCHL been tested for temperature regulation at the level of gene expression, protein accumulation and protein distribution?.

Response:

- This is a very good point. Temperature-regulation of PCH1 and/or PCHL could either increase or decrease the direct effect of temperature on phyB dark reversion. We plan to

investigate this in more detail but think that including such data in the present manuscript is premature and beyond the scope of the manuscript.

Reviewer Comment:

3) Authors state that the effect of PHC1 and PHCL over phyB dependent photomorphogenesis is not through regulation of phyB protein levels. However, SF4 protein blot may show slight differences of phyB in the dark. Quantification and replicas of the experiment would be required to clarify that no phyB protein changes are involved in *phc1* and *phcl* phenotypes.

Response:

- We fully agree that this is an important point and therefore we repeated the experiment and quantified the western blot signals (Supplementary Fig. 5). To assess the reliability of the quantification we loaded the same Col-0 extract in two different lanes (second and last lanes in panel B and C); quantification of the bands (normalised to the actin bands) revealed differences in the range of 15% between the two lanes, in which equal amounts of the same Col-0 extract were loaded. PhyB levels in dark-grown *pch1*, *pchl*, and *pch1 pchl* seedlings are within the range of wild type levels $\pm 15\%$, suggesting that phyB levels in the dark are comparable in wild type, *pch1*, *pchl*, and *pch1 pchl* (as also suggested by reviewer #2, comment 4: "...western immunoblot assays indicate that phyB levels in Col-0 and *pch1 pchl* mutant are comparable ..."). Thus, we conclude that the effect of PCH1 and PCHL on phyB responses is not through regulation of phyB protein abundance.

Reviewer Comment:

4) Figure 2F shows an apparent difference of phyB-GFP abundance in red light in the absence of *phc1 pchl*. Is this due to the optical plane used? Or are there differences in the abundance of phyB in R-light?

Response:

- The western blot data in Supplementary Fig. 5 show that the endogenous phyB levels are very similar in wild type and *pch1 pchl* seedlings. Even though the settings for image acquisition and image processing was identical for all pictures in Fig. 2F, it is hard to make any conclusions regarding phyB-GFP levels because the localisation of phyB-GFP differs depending on incubation time in the dark and presence/absence of PCH1/PCHL. The localisation and signal intensity of phyB-GFP after 12 h incubation in R (Fig. 2F, very left) is very similar in wild type and the *pch1 pchl* background, suggesting that the phyB-GFP levels are not controlled by PCH1/PCHL. In contrast, the phyB-GFP levels after 12 h incubation in R followed by 4 h in the dark cannot be compared because the localisation of phyB-GFP is very different depending on the presence/absence of PCH1/PCHL. However, data shown in Fig. 3B suggest that the phyB-GFP levels do not change during incubation in the dark following exposure to R (also see response to comment 4 by reviewer 2).

Reviewer Comment:

5) In Figure 4, authors show evidence of PHC1 and PHCL FR and B -light regulation at the level of gene expression. However, a missing piece of information is whether these changes in gene expression translate in changes in protein content and modulation of phyB-dark reversion. Can the authors complement their data with evidence of changes in protein content/distribution under the mentioned light regimes?

Response:

- Western blot data in Fig. 4A show that in seedlings expressing HA-YFP-tagged PCH1 under the control of the endogenous *PCH1* promoter HA-YFP-PCH1 is barely detectable in the dark but accumulates to detectable levels upon 8 hours irradiation with FR, B, or R. The increase in HA-YFP-PCH1 levels upon irradiation with FR, B, or R for 8 hours can also be detected by fluorescence microscopy as shown in Fig. 4B. Thus, we conclude that endogenous PCH1 protein levels are upregulated under the conditions used for pre-irradiation in Fig. 4E and F. We do not have any lines expressing tagged PCHL under the control of the endogenous *PCHL* promoter and PCHL antibodies are not available. Therefore, we cannot test accumulation of the endogenous PCHL protein at the moment. However, Fig. 4E and F show that both PCH1 and PCHL are required for responsiveness amplification, suggesting that not only PCHL transcript but also protein levels are increased upon pre-irradiation with FR or B.
- We fully agree with the reviewer that it would be interesting to measure phyB dark reversion following 8 hours irradiation with FR, B, or R. However, this is technically impossible for several reasons. First, the 8 h pre-treatment with light results in accumulation of chlorophyll/chlorophyll precursors, which interfere with ratiospect measurements. Second, we used lines expressing wild type levels of phyB for this experiment, which cannot be detected in ratiospect measurements; only over-expressed phyB can be measured. Third, measuring phyB in phyB over-expressing lines would not be possible with the experimental setup in Fig. 4E and F. Measuring phyB requires *phyA* mutant background or degradation of *phyA* by exposure to red light, since *phyA* interferes with measuring phyB. However, the experiment requires *phyA*, which is essential for perceiving the 8 h light pre-treatment, and applying an extended red light treatment to trigger *phyA* degradation is not compatible with the experiment (i.e. applying only a 5 min red light pulse).

Reviewer Comment:

6) In their analyses of PHC1 and PHCL expression under R, B and FR(Figure S7) there are considerable differences in the kinetics and light sensitivity of PHC1 and PHCL expression. PHCL expression seems to be quite low under R regimes compared to B or FR, however the contribution to the phenotypes in Figures 10 and Figure 2 are similar.

Can the authors comment on this observation? Is this regulation related to differential roles of PHC1 and PHCL?

Response:

- We fully agree with the reviewer that data in Supplementary Fig. 8 (= Supplementary Fig. 7 in the old version of the manuscript) suggest that there are differences between *PCH1* and *PCHL* expression regarding kinetics and light sensitivity. Both PCH1 and PCHL proteins bind to phyB and both reduce phyB dark reversion, suggesting that the PCH1 and PCHL proteins have similar function. Thus, the primary difference between *PCH1* and *PCHL* might be the regulation and there could be conditions under which primarily *PCH1* regulates phyB dark reversion while under other conditions *PCHL* may be more important. This idea is supported by the finding that *PCH1* and *PCHL* are both required for the responsiveness amplification while *PCH1* is more important than *PCHL* for regulation of phyB dark reversion after light off in seedlings grown in light/dark cycles.

Reviewer Comment:

7) In the same figure (S7), there are differences in the expression patterns of the 3 replicas provided. How does the average expression looks like? Are there kinetic differences on the expression of PHC1 and PHCL in response to light quality? (Fig S7).

Response:

- In the revised version of the manuscript we include panels showing average expression (Supplementary Figs. 8D, 8H, 10D, 10H, 11D, and 11H). Moreover, we also show average expression in Fig. 4C and D in the revised version of the manuscript. However, we think it is worth to also include data for the individual replicates in Supplementary Figs. 8, 10, and 11. These show that there is some variation but that the overall expression pattern is very similar between the replicates.
- The data in Supplementary Fig. 8 (= Supplementary Fig. 7 in the old version of the manuscript) suggest that upregulation of *PCH1* and *PCHL* in red light is transient with a peak after 1 or 3 hours and that expression returns to basal levels after 12 hours (as discussed later, this effect might be due to rapid degradation of phyA in red light). In contrast, blue light and in particular far-red light result in more sustained upregulation of *PCH1* and *PCHL* transcript levels, which is also confirmed by GUS stains of *PCH1*- and *PCHL*-promoter:GUS reporter lines (new data; presented in Supplementary Fig. 9).

Reviewer Comment:

8) Does the observed and FR light modulation of PHCL and PHC1 result in altered expression of phyB-signaling components? (i.e. PIFs, others?)

Is there an interaction of PHCI/L with phyA? ,and impact on phyA signaling under tested FR conditions?

Response:

- This is a very good point, which however is not easy to address. For several reasons we think that simple expression analyses (qRT-PCR, Western blot) would not give conclusive results or even might give misleading results. First, testing expression only for a subset of components involved in phyB signalling would be misleading, since it is possible that the expression of these components is not altered in *pch1 pchl* while the expression of components that have not been tested might be altered. Thus, to obtain unbiased data we would have to do a full microarray/RNAseq experiment for wild type vs. *pch1 pchl*/compare the full proteome of wild type and *pch1 pchl*, which is beyond the scope of this manuscript. Second, testing expression of phyB signalling components would only make sense if these components were absolutely specific for phyB and not also functioning downstream of phyA. Otherwise, upregulation of these components by the FR pre-treatment would also enhance signalling through phyA activated by the red light pulse and not only through phyB. However, both COP1/SPA and PIFs work downstream of phyA and phyB. Even PIF4 and PIF5 that do not interact with phyA play a role not only downstream of phyB but also phyA (Lorrain et al., 2009, Plant J., 60, 449–461). In summary, it is well established that the rate of phyB dark reversion is a key factor that determines the response to red light pulses and we have shown that PCH1 and PCHL regulate the dark reversion of phyB. Thus, the model shown in Fig. 5 is well supported by data. However, we agree that we cannot rule out that PCH1/PCHL – in addition to reducing phyB dark reversion – also could affect phyB downstream signalling by other (still hypothetical) mechanisms. We mention this possibility in the revised version of the manuscript (lines 224ff).
- In the co-IP/mass spec approach by Huang and co-workers (Huang et al., 2016, eLife, 5, 1–27) phyA co-purified with PCH1 in wild type and *phyB-9* background. Thus, phyA associates with a PCH1 containing complex and this association does not require phyB. Consistent with these data we observed in co-IP assays from stable transgenic plants that phyA binds – directly or indirectly – to PCH1 and PCHL (data not included in the manuscript). However, using the C-terminal half of phyA for Y2H assays, Huang and co-workers did not find any interaction with PCH1.

- The *pch1*, *pchl*, and *pch1 pchl* mutants were indistinguishable from the wild type under the far-red light conditions used for the assays shown in Supplementary Fig. 3H (= Supplementary Fig. 2H in the old version of the manuscript); moreover, hypocotyl growth in far-red light was similar in PCH1ox, PCHLox, and the wild type (Supplementary Fig. 3G).

Reviewer Comment:

9) PHC1 has been reported by other authors as an F-box protein, how would the potential function of PHC1/PHCL relate their involvement in dark-reversion?

Response:

- There is only weak support for the existence of the PCH1 splicing variants containing the F-box (see response to comment 1 for details). In addition, Huang and co-workers (Huang et al., 2016, Mol. Cell. Proteomics, 15, 201–217) did not identify any peptides mapping to the F-box containing domain of PCH1 in their co-IP/mass spec approach, suggesting that the F-box containing variant is not expressed under the conditions used for the experiment (or is not expressed at all); also in our own co-IP/mass spec approach we could not identify the F-box containing PCH1 version. Furthermore, we could never amplify the F-box containing splicing variant of PCH1 from total cDNA and therefore we used the PCH1 At2g16365.2 splicing variant for all experiments. Huang and co-workers also used the PCH1 At2g16365.2 splicing variant (i.e. the splicing variant not containing the F-box) for their study (Huang et al., 2016, eLife, 5, 1–27). In conclusion, PCH1 has an effect on phyB dark reversion under conditions where the F-box containing splice variant cannot be detected and therefore we conclude that the F-box is not required for PCH1 mediated regulation of phyB dark reversion. This view is also supported by the fact that also PCHL regulates phyB dark reversion but the only PCHL splicing variant does not contain an F-box.

RESPONSE TO REVIEWER 2:

Reviewer Comment:

(1) Fig. 4A of the immunoblot assay, please add “R” control for comparison to FR and B light induction of HA-YFP-PCH1. It is suggestive to analyze PCHL protein levels in response to light treatments as well. Fig. 4B and S7A~S7C indicates ~3-fold difference in PCH1 transcript levels between dark and FR conditions, how come the difference in protein levels as shown in Fig. 4A and 4B is more than 3-fold (evidently detectable vs below detection threshold). In Fig. S7, two out of three biological replicates show that under R conditions, PCH1 is only transiently upregulated 1h after R exposure. It is well known that phyA is rapidly degraded upon R exposure; but within 1h, phyA is degraded by only ~50%; and by 3h, it is almost gone. Collectively, this suggests that R-induced PCH1 transcript upregulation is mediated by phyA before degradation. Such an interpretation is supported by Fig. 4C-D: in the phyA mutant, the PCH1/PCHL transcript level has no response to 1h R exposure. The conclusion drawn in the first paragraph of page 9 is thus well supported, but can be discussed more for its implication.

Response:

- We added the “R” control for comparison (Fig. 4A) and also analysed HA-YFP-PCH1 accumulation in response to R by fluorescence microscopy (Fig. 4B). These experiments show that HA-YFP-PCH1 protein levels are also upregulated by R pre-treatment, which is consistent with the phyA dependent upregulation of PCH1 transcript levels in response to R (Fig. 4C; Supplementary Fig. 10A–D).

- We do not have any PCHL antibodies and we do not have any lines expressing PCHL under the control of the endogenous promoter. Thus, we currently cannot analyse PCHL protein levels. However, we would like to point out that we have analysed PCH1 protein levels and that PCH1 is the dominant member of the PCH1/PCHL pair.
- We agree that the upregulation of PCH1 protein levels in response to FR might be more than 3-fold, though it is not possible to quantify the upregulation based on the western blot shown in Fig. 4A. However, we do not exclude that the PCH1 protein might be stabilised in light or destabilised in the dark, which would explain a more prominent regulation of protein than transcript abundance.
- We fully agree with the reviewer that the transient upregulation of PCH1/PCHL can be explained by degradation of phyA. We discussed it in more detail in the manuscript (lines 176ff).

Reviewer Comment:

(2) It was not mentioned in the main text, but Fig 2A and 2B clearly show that in the dark PCH1ox and PCHLox are shorter than Col-0, phyB-9 and phyA-211phyB-9. Is it due to later germination of the transgenic lines, caused by the growth assay method (on filter paper plus water for growth; no MS salts), or indeed indicate the elongation suppression role of overexpressed PCH1/PCHL? Furthermore, the phenotype of PCH1/PCHLox causing hypersensitivity to Rp treatment is very striking. This observation came from growth assay on filter paper plus pure water. I would ask the authors to repeat the growth results on MS medium.

Response:

- We repeated the experiments shown in Fig. 2A and B on 0.5× MS medium. The response of PCH1ox and PCHLox seedlings to red light pulses was even more pronounced on 0.5× MS medium than on water. The comparison of seedlings grown on water and on 0.5× MS medium is shown in Supplementary Fig. 4C–F. Furthermore, the dark phenotype of PCH1ox and PCHLox seedlings was weaker on 0.5× MS medium than on water but even on 0.5× MS medium dark-grown PCH1ox and PCHLox seedlings are slightly shorter than the wild type. We did not observe delayed germination of PCH1ox/PCHLox seeds, suggesting that overexpressed PCH1/PCHL might slightly suppress hypocotyl elongation in the dark. However, we would like to point out that even PCH1ox/PCHLox seedlings are fully etiolated in the dark. We added a sentence in the revised version of the manuscript to mention the growth phenotype of dark-grown PCH1ox/PCHLox seedlings (line 88ff).

Reviewer Comment:

(3) In Fig S2, why the fluence rate response trends of Col-0 in panel D and F are so different from panels A~C, F and I, very minor change in hypocotyl length even under $\geq 20\mu\text{mol m}^{-2} \text{s}^{-1}$ R. This is also different from most, if not all, reported Col-0 hypocotyl growth under continuous R. As such, the data of other genotypes in panels D-E are not so trustable. Please repeat the experiments.

Response:

- We thank the reviewer for pointing out this issue. We investigated this issue carefully and noticed that the temperature settings have been changed inadvertently between the two sets of experiments (22 °C for experiments shown in A–C, F, and I vs. slightly more than 22 °C for experiments shown in D and E). Such temperature increase could be causative for the weaker response to red light. We repeated the experiments shown in D and E at 22 °C; in the revised version of Supplementary Fig. 3 (= Supplementary Fig. 2 in the old version of the manuscript) hypocotyl growth of seedlings in D and E is similar to A–C, F, and I.

Reviewer Comment:

(4) Fig. S4 western immunoblot assays indicate that phyB levels in Col-0 and *pch1 pchl* mutant are comparable, why total phyB measured in Fig. 3B by ratiospect is ~2-fold difference between Col-0 and the mutant, given that the same phyB-GFP OE line was used?

Response:

- We fully agree with reviewer #2 that phyB levels in Col-0 and *pch1 pchl* mutant background are comparable, which is further confirmed by additional immunoblot data and quantification of phyB levels presented in the revised version of the manuscript (Supplementary Fig. 5). The *pch1 pchl* phyB-GFP OE line was obtained by crossing phyB-GFP OE into *pch1 pchl* background and therefore one would not expect a 2-fold difference regarding phyB-GFP levels. However, for unknown reasons, overexpression of phyB-GFP often results in full or partial silencing of the phyB-GFP transgene. In some phyB-GFP transgenic lines, silencing occurs already in the T1 generation, while other lines initially overexpress phyB-GFP but gradually silence the transgene in later generations. Thus, the likely reason for the lower levels of phyB-GFP in *pch1 pchl* is partial silencing of the transgene. However, we would like to point out that the kinetics of phyB dark reversion does not depend on the phyB concentration and therefore data presented in Fig. 3 show that overexpressed and endogenous PCH1 and PCHL delay phyB dark reversion.

Reviewer Comment:

(5) For Fig. S5, please provide schematic illustration of the PCH1 and PCHL gene models, which should also include the T-DNA insertion sites and locations/orientations of primers used for transcript detection. I wonder if the primer pair used for the PCH1 transcript assay spans the T-DNA insertion site so that no transcript can be detected in the mutant.

Response:

- We added an additional panel to Supplementary Fig. 6 (= Supplementary Fig. 5 in the old version of the manuscript) in the revised manuscript showing the *PCH1/PCHL* gene models, the T-DNA insertion sites, and the PCR products amplified in qRT-PCR analyses (Fig. 4C and D; Supplementary Figs. 6B, 8, 10, and 11). The reverse primer used for detection of *PCH1* transcript levels spans the exon 1/2 junction, which is downstream of the T-DNA insertion, while the primer pair used for detection of the *PCHL* transcript spans a region upstream of the T-DNA insertion site. This might explain why residual expression of (most likely truncated) *PCHL* can be detected in *pchl*, while no *PCH1* transcript is present in *pch1*.

Reviewer Comment:

(6) Fig. 1C, the immunoblot band intensities for the input proteins are so faint compared to the IP bands and input bands in Fig. 1D. Could the authors improve the blot image quality?

Response:

- We improved the image quality by adjusting brightness and contrast.

Reviewer Comment:

(7) The methods section is quite lengthy, especially about vectors and cloning, and plant materials. Meanwhile, the key information about the sequence lengths of promoters being cloned is not reported. The very detailed cloning methods as well as primer list can be put into supplemental data.

Response:

- We agree that the section on cloning of constructs as well as the primer list can be moved to the Supplementary Information; this has been done in the revised version of the manuscript. Based on the genomic sequence of *PCH1/PCHL/PHYB* and the sequence of the primers used for cloning of the respective construct it is possible to derive the sequence and length of the promoters. However, we fully agree with the reviewer that it is appropriate to explicitly mention the length of the promoters as we do in the Supplementary Methods section of the revised version of the manuscript (lines 235, 249, 264f).

Reviewer Comment:

Minor points:

(1) Ref 45 misses the publication year.

(2) Page 19, line 381. Is the custom a-phyB antibody used for co-IP B6-B3, same as that used for immunoblot analysis (line 395)?

(3) Typo in Fig S1 legend, line 33; please swap “a-c-Myc” and “a-GFP” antibodies. Typo in Fig S10 legend, Line 137, “see Figures 4A and 4B for...” should be “see Figures 4E and 4F...”.

Response:

- We added the publication year and corrected the typos.
- The antibody used for the immunoblot analysis and the co-IP is the same (B6-B3; Hirschfeld et al., 1998, *Genetics*, 149, 523–535); we inserted this information in the revised version of the manuscript (line 358)

Reviewers' comments:

Reviewer #1 (Remarks to the Author):

Authors have fully addressed and incorporated the modifications suggested or addressed comments raised in previous version. Therefore, I would support publication.

Reviewer #2 (Remarks to the Author):

The authors have thoroughly and satisfactorily addressed almost all comments raised in the first review cycle. Here are three more comments about the revised MS and new results.

(1) In regard to the ~ 2-fold difference in the phyB-GFP levels between WT and pch1 pchl mutant as determined by ratiospect (Fig. 3B), the partial gene silencing explanation is not convincing. It is notorious that phyB OE frequently induces gene silencing, consequently conferring transgenic plants of phyB mutant-like phenotypes. On the other hand, stable phyB-OE transgenics seldom exhibit slight gene silencing effect over generations. In short, to support the authors' claim, a western blotting assay is expected to clarify this difference; the ~2-fold difference in protein levels should be easily detected.

(2) For the newly added promoter::GUS assay results, did the authors intentionally omit the important chemicals ferri/ferrocyanide in the staining solution (lines 327~328), and understand the risk of such omission in expanding GUS staining regions?

(3) In the new Supplementary Figure 2, please make the sequence names concise and informative so readers can be inferred each protein sequence is from which species/genus. In addition, the alignment clearly shows that PCHL is a truncation or deletion version of the PCH1-like gene family, missing a stretch of highly conserved residues in the middle region. The authors should discuss the implication of such deletion on the weaker function of PCHL compared to PCH1.

RESPONSE TO REVIEWER 2:

Reviewer Comment:

(1) In regard to the ~2-fold difference in the phyB-GFP levels between WT and *pch1 pchl* mutant as determined by ratiospect (Fig. 3B), the partial gene silencing explanation is not convincing. It is notorious that phyB OE frequently induces gene silencing, consequently conferring transgenic plants of phyB mutant-like phenotypes. On the other hand, stable phyB-OE transgenics seldom exhibit slight gene silencing effect over generations. In short, to support the authors' claim, a western blotting assay is expected to clarify this difference; the ~2-fold difference in protein levels should be easily detected.

Response:

- To generate the line expressing phyB-GFP in *pch1 pchl* background we crossed *phyA phyB* phyB-GFP into *pch1 pchl*. In the F2 generation we selected for plants that are homozygous for *phyA* and the phyB-GFP transgene. Finally, different F3 lines homozygous for *phyA* and the phyB-GFP transgene were used for detection of phyB by ratiospect; from these F3 lines we selected the line producing the highest signal in the ratiospect for further experiments. This line contains the endogenous phyB in addition to the transgene, which was not correctly stated in the Material and Method section due to a communication error. In the revised version of the manuscript we indicate in the Material and Method section as well as in the legend to Figure 3 that endogenous phyB is still present in the *phyA pch1 pchl* phyB-GFP line.
- As requested by the reviewer, we quantified the phyB(-GFP) levels in *PCH1/PCHL* WT (*phyA phyB* phyB-GFP) and *pch1 pchl* mutant background (*phyA pch1 pchl* phyB-GFP). The quantification shows that the total level of phyB (i.e. endogenous phyB and phyB-GFP) in *phyA pch1 pchl* is roughly half of the total phyB level (i.e. phyB-GFP) in *phyA phyB* phyB-GFP. This is fully consistent with the results of the quantification of total phyB levels by ratiospect (Figure 3). Furthermore, the Western blot quantification (Supplementary Fig. 8 in the revised version of the manuscript) shows that endogenous phyB accounts for roughly one third of the total phyB levels in *phyA pch1 pchl* phyB-GFP, which contributes to the detectability of phyB in this line by ratiospect measurements.
- We have shown that endogenous phyB levels are not regulated by PCH1 and PCHL (Supplementary Figure 5), upon which also reviewer #2 agreed (reviewer comments on the previous version of the manuscript). Thus, it appears unlikely that the absence of PCH1/PCHL in *phyA pch1 pchl* phyB-GFP is the reason for the lower levels of phyB-GFP compared to *phyA phyB* phyB-GFP. We hypothesised that “partial silencing” might be responsible for this. We agree with the reviewer – and we also observed this in our own work – that in many cases phyB transgenes are either fully expressed or they are fully silenced. However, for the *phyA phyB* phyB-GFP line we observed that the hypocotyl length gradually increased in successive generations (i.e. T4 has longer hypocotyls than T3, which has longer hypocotyls than T2), suggesting that in this line the phyB-GFP levels decrease progressively and not in an all-or-non fashion. This behaviour – progressive decrease of expression levels – might be uncommon but there are examples described in the literature (Mittelstein Scheid, 1995; Cannell et al., 1999; Hagan et al., 2003). We therefore hypothesised that the different phyB-GFP levels in *phyA phyB* phyB-GFP (T3 generation)

and *phyA pch1 pchl* phyB-GFP (T6 generation with respect to transgene) might be due to suppression of transgene expression but do not exclude other explanations. It should also be noted that *pch1* and *pchl* are SALK T-DNA insertion lines that contain the 35S promoter on the T-DNA used for mutagenesis (Alonso et al., 2003; Baulcombe et al., 1986) and that the phyB-GFP transgene crossed into the *pch1 pchl* background is expressed under the control of the 35S promoter. It has been shown that SALK lines can induce transcriptional gene silencing of transgenes located on an independent T-DNA but also expressed from the 35S promoter (Daxinger et al., 2008; Mlotshwa et al., 2010). Thus the “SALK background” of the *pch1 pchl* mutant might also play a role in suppressing the expression of the p35S:phyB-GFP transgene in *pch1 pchl*.

- In summary, we would like to point out that the kinetics of phyB dark reversion does not depend on the phyB concentration and therefore data presented in Figure 3 show that overexpressed and endogenous PCH1 and PCHL delay phyB dark reversion.

Reviewer Comment:

(2) For the newly added promoter::GUS assay results, did the authors intentionally omit the important chemicals ferri/ferrocyanide in the staining solution (lines 327~328), and understand the risk of such omission in expanding GUS staining regions?

Response:

- Including ferri/ferrocyanide in the staining buffer prevents diffusion of the primary reaction product of GUS but also reduces the GUS enzyme activity (and thereby the signal strength). To obtain a well detectable signal we did not include ferri/ferrocyanide in the staining buffer. We obtained very specific staining patterns, e.g. in root tips, suggesting that diffusion of the reaction product was not a problem. However, we agree with the reviewer that this is an important issue and therefore we repeated the experiment using staining buffer supplemented with ferri/ferrocyanide. To detect GUS staining – in particular for pPCHL:GUS – we had to increase the incubation time but the Col-0 control confirms that no unspecific staining occurred. The staining pattern with and without ferri/ferrocyanide is very similar. However, to be in line with the commonly used protocol for GUS stains we now show the results of the GUS stain including ferri/ferrocyanide in Supplementary Fig. 10 (Supplementary Fig. 9 in the previous version of the manuscript).

Reviewer Comment:

(3) In the new Supplementary Figure 2, please make the sequence names concise and informative so readers can be inferred each protein sequence is from which species/genus. In addition, the alignment clearly shows that PCHL is a truncation or deletion version of the PCH1-like gene family, missing a stretch of highly conserved residues in the middle region. The authors should discuss the implication of such deletion on the weaker function of PCHL compared to PCH1.

Response:

- In the revised version of Supplementary Figure 2, the sequences are labelled by the full species name and the Uniprot sequence identifier.

- PCHL lacks a motif that is highly conserved in PCH1 and PCH1/PCHL related sequences from other species, which is interesting and which might explain the partially different and/or weaker function of PCHL compared to PCH1. We discuss this aspect in the revised version of the manuscript (fourth paragraph in the discussion).

Alonso, J.M. et al. (2003). Genome-wide insertional mutagenesis of *Arabidopsis thaliana*. *Science* **301**: 653–657.

Baulcombe, D.C., Saunders, G.R., Bevan, M.W., Mayo, M.A., and Harrison, B.D. (1986). Expression of biologically active viral satellite RNA from the nuclear genome of transformed plants. *Nature* **321**: 446–449.

Cannell, M.E., Doherty, A., Lazzeri, P.A., and Barcelo, P. (1999). A population of wheat and tritordeum transformants showing a high degree of marker gene stability and heritability. *Theor. Appl. Genet.* **99**: 772–784.

Daxinger, L., Hunter, B., Sheikh, M., Jauvion, V., Gascioli, V., Vaucheret, H., Matzke, M., and Furner, I. (2008). Unexpected silencing effects from T-DNA tags in *Arabidopsis*. *Trends Plant Sci.* **13**: 4–6.

Hagan, N.D., Spencer, D., Moore, A.E., and Higgins, T.J.V. (2003). Changes in methylation during progressive transcriptional silencing in transgenic subterranean clover: Methylation changes during progressive transcriptional silencing. *Plant Biotechnol. J.* **1**: 479–490.

Mlotshwa, S., Pruss, G.J., Gao, Z., Mgutshini, N.L., Li, J., Chen, X., Bowman, L.H., and Vance, V. (2010). Transcriptional silencing induced by *Arabidopsis* T-DNA mutants is associated with 35S promoter siRNAs and requires genes involved in siRNA-mediated chromatin silencing. *Plant J.* **64**: 699–704.

Mittelstein Scheid, O. (1995). Transgene Inactivation in *Arabidopsis thaliana*. In *Gene Silencing in Higher Plants and Related Phenomena in Other Eukaryotes*, P. Meyer, ed (Springer Berlin Heidelberg: Berlin, Heidelberg), pp. 29–42.